# Association between Antiretroviral Therapy and Cancers among Children Living with HIV in Sub-Saharan Africa

**DOI:** 10.3390/cancers13061379

**Published:** 2021-03-18

**Authors:** Heather Haq, Peter Elyanu, Shaun Bulsara, Jason M. Bacha, Liane R. Campbell, Nader K. El-Mallawany, Elizabeth M. Keating, Grace P. Kisitu, Parth S. Mehta, Chris A. Rees, Jeremy S. Slone, Adeodata R. Kekitiinwa, Mogomotsi Matshaba, Michael B. Mizwa, Lumumba Mwita, Gordon E. Schutze, Sebastian R. Wanless, Michael E. Scheurer, Joseph Lubega

**Affiliations:** 1Department of Pediatrics, Baylor College of Medicine, Houston, TX 77030, USA; heather.haq@bcm.edu (H.H.); shaun.bulsara@bcm.edu (S.B.); bacha@bcm.edu (J.M.B.); lianec@bcm.edu (L.R.C.); nader.el-mallawany@bcm.edu (N.K.E.-M.); elizabeth.keating@hsc.utah.edu (E.M.K.); psmehta@texaschildrens.org (P.S.M.); chris.rees@childrens.harvard.edu (C.A.R.); jsslone@texaschildrens.org (J.S.S.); mmizwa@bcm.edu (M.B.M.); schutze@bcm.edu (G.E.S.); scheurer@bcm.edu (M.E.S.); 2Baylor College of Medicine International Pediatric AIDS Initiative (BIPAI) at Texas Children’s Hospital, Houston, TX 77030, USA; zakaldovan@gmail.com; 3Baylor College of Medicine Children’s Foundation-Uganda, Kampala, Uganda; pelyanu@baylor-uganda.org (P.E.); gkisitu@baylor-uganda.org (G.P.K.); akekitiinwa@baylor-uganda.org (A.R.K.); 4Baylor College of Medicine Children’s Foundation-Tanzania, Mbeya, Tanzania; lmwita@baylortanzania.or.tz; 5Baylor College of Medicine Children’s Foundation-Lesotho, Maseru, Lesotho; 6Global Hematology Oncology Pediatric Excellence Program, Texas Children’s Cancer and Hematology Centers, Houston, TX 77030, USA; 7Baylor College of Medicine Children’s Foundation-Malawi, Lilongwe, Malawi; 8Botswana-Baylor Children’s Clinical Centre of Excellence, Gabarone, Botswana; matshaba@bcm.edu

**Keywords:** pediatric, HIV-associated cancers, Kaposi sarcoma, lymphoma, antiretroviral therapy, sub-Sahara Africa

## Abstract

**Simple Summary:**

Most children infected with HIV live in Sub-Sahara Africa (SSA). These children are at risk of cancers related to HIV infection, but the degree of this risk and how it is influenced by antiretroviral therapy (ART) is unknown. In this study, we determined the subtypes, incidence, and risk factors of cancers in children with HIV in SSA and receiving ART with the goal of learning how we may prevent these cancers. We found that Kaposi sarcoma and lymphoma are the most common, comprising about 77% and 19% of cancers in these children, respectively. For every 100,000 person-years, 47.6 children developed cancer. Waiting to start ART until after 2 years old and having had severe immunosuppression were the two biggest risk factors for cancer that we identified. The findings justify the recommendations to start children on ART as soon as they are diagnosed with HIV regardless of their CD4 immune status.

**Abstract:**

Approximately 91% of the world’s children living with HIV (CLWH) are in sub-Saharan Africa (SSA). Living with HIV confers a risk of developing HIV-associated cancers. To determine the incidence and risk factors for cancer among CLWH, we conducted a nested case-control study of children 0–18 years from 2004–2014 at five centers in four SSA countries. Incident cases of cancer and HIV were frequency-matched to controls with HIV and no cancer. We calculated the incidence density by cancer type, logistic regression, and relative risk to evaluate risk factors of cancer. The adjusted incidence density of all cancers, Kaposi sarcoma, and lymphoma were 47.6, 36.6, and 8.94 per 100,000 person-years, respectively. Delayed ART until after 2 years of age was associated with cancer (OR = 2.71, 95% CI 1.51, 4.89) even after adjusting for World Health Organization clinical stage at the time of enrolment for HIV care (OR = 2.85, 95% CI 1.57, 5.13). The relative risk of cancer associated with severe CD4 suppression was 6.19 (*p* = 0.0002), 2.33 (*p* = 0.0042), and 1.77 (*p* = 0.0305) at 1, 5, and 10 years of ART, respectively. The study demonstrates the high risk of cancers in CLWH and the potential benefit of reducing this risk by the early initiation of ART.

## 1. Introduction

An estimated 1.7 million children under 15 years old are infected with human immunodeficiency virus (HIV) worldwide, 91% of them in sub-Saharan Africa (SSA) [1]. With improved access to antiretroviral therapy (ART) and the adequate treatment of opportunistic infections in recent decades, the survival of children with HIV infection in SSA has improved significantly. Accordingly, the relative significance of non-infectious complications of HIV infection including pediatric HIV-associated cancers as causes of morbidity and mortality in children living with HIV (CLWH) has increased. Kaposi sarcoma (KS) and lymphoma are the most common pediatric HIV-associated cancers [2,3,4,5].

Only a small proportion of CLWH in the United States develop cancer [6]. The predisposing factors for cancer in the context of pediatric HIV infection remain unclear. Hence, there are no precise, evidence-based clinical strategies for primary prevention and early detection of pediatric HIV-associated cancers [7]. Possible contributions to the increased risk of cancers in CLWH may include immune dysregulation, uncontrolled HIV replication, impaired tumor immune surveillance, and a high incidence of co-infection with other oncogenic viruses such as Epstein-Barr virus (EBV) and Human herpes virus type 8 (HHV8) [8,9]. There is also some evidence that HIV proteins or HIV viremia directly contribute to oncogenesis, regardless of host immune status [10].

Among adults living with HIV, antiretroviral therapy has been shown to alter the natural history and phenotypes of HIV-associated cancers [11,12]. The incidence of KS and non-Hodgkin lymphoma (NHL) decreases about 3-fold and 2-fold, respectively, in adults on ART [13,14,15]. However, the risk of HIV-associated cancers remains higher in adults with HIV than in HIV negative adults, even after prolonged use of ART, suppression of HIV viral load, and restoration of CD4 lymphocyte counts [16]. Furthermore, some studies in adults have found that immune reconstitution is associated with increased risk of occurrence and mortality from KS, NHL, and Hodgkin lymphoma after commencing ART [13,17,18,19,20]. The biological factors that underlie HIV-associated oncogenesis after prolonged, effective ART are not clear. Specific ART agents such as protease inhibitors and others may have specific properties that repress tumorigenesis [21]. However, it is plausible that the effects of HIV on host immune homeostasis and other oncogenic pathways may be permanent. Therefore, the timing of ART initiation regardless of immune function and clinical complications may be critical to prevent the future development of cancer. On the other hand, there is a putative risk of specific ART agents, such as nucleoside or non-nucleoside analogs, inducing malignant mutagenesis; this has been shown in in vitro studies and animal models, but has not been examined in clinical studies [22].

This study aimed to determine the incidence and risk factors associated with cancer in CLWH in four countries in SSA by utilizing data from five pediatric HIV treatment centers. These HIV clinical centers of excellence are part of Baylor College of Medicine International Pediatric AIDS Initiative (BIPAI) at Texas Children’s Hospital, a large multi-national network for pediatric HIV care and treatment with clinical centers of excellence in six countries in SSA [23]. A further understanding of the modifiable risk factors of cancers among CLWH may allow for interventions to reduce the incidence of cancers, and, hence, mortality in CLWH in SSA.

## 2. Materials and Methods

### 2.1. Study Design and Study Population

We conducted a nested case-control study using data from an observational clinical cohort of CLWH aged 0–18 years who enrolled in HIV care between January 2004 and December 2014 at five BIPAI centers in four SSA countries: Botswana (Gaborone), Malawi (Lilongwe), Tanzania (Mwanza and Mbeya), and Uganda (Kampala). Following diagnosis with HIV, children in the clinical cohort were initiated on ART and provided comprehensive HIV care and treatment following respective country guidelines. This study population is representative of children with HIV in multiple regions in SSA, both in the early ART era (~2004–2008) when children typically presented with symptomatic HIV and AIDS-defining diseases and were started on ART after manifesting severe immunosuppression, and the latter ART era (~2009–2014) when children were typically diagnosed with HIV through early infant screening and diagnosis and started on ART as soon as a diagnosis of HIV is made. Children enrolled in care were followed longitudinally with routine and sick clinic visits. During follow-up visits, children were reviewed for the presence of any complications including cancers. Clinical data for these patients were managed using a standardized electronic medical record (EMR) system.

### 2.2. Cases

Cases were defined as CLWH aged 0–18 years and diagnosed with any cancer. Cases were identified through searching BIPAI clinics’ EMR for the following terms: “cancer”, “malignancy”, “neoplasm”, “tumor”, “Kaposi”, “sarcoma”, “lymphoma”, “Burkitt”, and “Hodgkin”. Cancer diagnoses were confirmed by reviewing histopathology reports when available. Diagnoses were annotated by the most specific pathology diagnosis on the report. For example, whereas some children had a very specific type of lymphoma such as “Burkitt lymphoma”, others had a broader diagnosis such as “non-Hodgkin lymphoma”. Only incident cancer cases (defined for this study as at least three months after enrollment in HIV care) were included in the nested case-control analysis.

### 2.3. Controls

Controls were CLWH aged 0–18 years and without a diagnosis of cancer during the follow-up period. Controls were selected using incidence density sampling (ratio 1:5) and matched on follow-up time and sex.

### 2.4. Study Variables and Definition

The variables considered in the analysis of risk factors included: date of ART initiation, age at ART initiation, ART regimen(s), WHO HIV clinical stage at ART initiation, and CD4 lymphocyte nadir. CD4 counts were categorized into immune suppression levels for age according to WHO classification [24]. The primary outcomes were diagnosis of cancer including type. Other variables that were considered for adjustment in the multi-variable model, had to meet certain criteria: (1) clinical or biological importance; (2) data available; (3) association with cancer on univariate analysis. The site/country variable was eliminated because of low clinical and biological significance and risk of differential misclassification between countries due to differences in cancer suspicion or diagnosis capability. The sex variable did not meet the association criterion on univariate analysis. The variable for type of ARV regimen was eliminated because of very small sample size on protease inhibitors. The variable for time to CD4 normalization was deemed to have a high likelihood of misclassification bias because CD4 counts were not performed at regular and standardized/same intervals for all children.

### 2.5. Statistical Analysis

We used the SAS 9.4 statistical package (SAS Institute Inc., Cary, NC, USA) to perform all analyses in this study. We summarized the baseline characteristics of the cases and control groups using descriptive statistics. We calculated the incidence density and risk factors associated with cancer. Crude incidence density rates were adjusted to the World standard population to calculate World age-standardized rates per 100,000 person-years [25].

To evaluate the association of putative risk factors and cancer, we used conditional logistic regression to compute odds ratios (unadjusted and adjusted, where applicable). We plotted and compared incidence density curves for key risk factors to estimate the relative risk of cancer at 1, 5, and 10 years from enrolment into HIV care at the BIPAI center. We used Gray’s test to assess whether incidence density curves were the same or different at all time points. All hypothesis test computations included 95% confidence intervals (95% CI) to estimate the precision of associations and *p*-values to estimate the robustness of the statistical models, which were considered robust if *p*-value < 0.05.

Ethical approval was obtained from Baylor College of Medicine (codes H-25403, H-26616, H-27755, H-32491, H-32678) in addition to each of the relevant institutional review boards in Botswana, Malawi, Tanzania, and Uganda.

## 3. Results

### 3.1. Incidence of Cancer among CLWH Receiving ART

From January 2004 through December 2014, CLWH in care at BIPAI centers in Botswana, Malawi, Tanzania, and Uganda contributed 98,394 person-years of follow-up, and 427 children were diagnosed with cancer. There were 117 incident cases of cancer during the study period, accounting for crude incidence density of 119 (95% CI 97.4, 140.4) and a World age-standardized rate of 47.6 cancer cases per 100,000 person-years of CLWH. The demographic characteristics of incident cancer cases and controls in the cohort are summarized in Table 1. Histopathological diagnosis was ascertained for 69.9% of incident cases.

### 3.2. Subtypes of Cancer

Table 2 summarizes the distribution of subtypes of cancer in this cohort by prevalent versus incident cases in relation to enrollment into HIV care at BIPAI centers.

KS was the most incident cancer—in 76.92% (*n* = 90) of children with crude incidence density of 91.5 (95% CI 74.4, 112.5) and a World age-standardized rate of 36.6 KS cases per 100,000 person-years. There were 22 incident cases of lymphoma, of which 17 were NHL (14.53%) and 5 were Hodgkin lymphoma (4.27%), representing a crude incidence density of 17.3 NHL cases per 100,000 person-years, (95% CI 10.7, 27.8), and 5.1 Hodgkin lymphoma cases per 100,000 person-years, (95% CI 2.1, 12.2). The World age-standardized rate of lymphomas was 8.9 per 100,000 person-years. Leukemia accounted for only 0.85% (*n* = 1) of incident cases representing a World age-standardized rate of 0.4 cases per 100,000 person-years. Non-lymphoid solid tumors included brain tumor, soft tissue sarcoma, nasopharyngeal carcinoma, osteosarcoma and retinoblastoma. These tumors contributed 3.42% of all incident cases.

### 3.3. Risk Factors for Cancer in Children with HIV Infection

Table 3 summarizes the association of cancer and risk factors in children with HIV infection.

#### 3.3.1. Age at Initiation of ART and Cancer Risk

Delayed commencement of ART until after 2 years of age was associated with 2.71-fold higher odds of cancer compared to starting before or at 2 years of age, (95% CI 1.51, 4.89, *p* = 0.0009). When adjusted for WHO clinical stage at time of enrolment, the odds of developing cancer if ART was commenced after 2 years of age increased to 2.84, (95% CI 1.57, 5.13, *p* = 0.0006). The risk of cancer associated with delayed ART was also present after adjusting for CD4 nadir, (aOR = 2.51, 95% CI 1.38, 4.58, *p* = 0.0027). We also analyzed the incidence density function of cancer against timing of commencement of ART to examine how the risk associated with delayed commencement of ART changed over time. Figure 1 shows the incidence density curve of cancer by age at the commencement of ART.

The relative risk of cancer for children who started ART after 2 years of age was 4.28-fold higher (95% CI 1.19, 15.41, *p* = 0.021) in the first year after enrollment, waned over time to 1.61 (95% CI 0.85, 3.06, *p* = 0.141) by 5 years, and 0.87 (95% CI 0.50, 1.50, *p* = 0.607 by 10 years in the cohort.

#### 3.3.2. CD4 Nadir and Cancer Risk

CD4 lymphocyte nadir had a complex association with cancer in this cohort. Although there was a trend of increasing odds of cancer by severity of CD4 nadir, none passed the threshold for statistical significance in this cohort. However, temporal analysis of the risk of cancer by CD4 nadir revealed a 3.53-fold and 6.19-fold relative risk of cancer at 1 year for advanced and severe CD4 nadir, respectively, compared to normal CD4 nadir (Figure 2).

The risk of cancer waned to insignificance by 5 years for the advanced CD4 nadir category. However, for the severe CD4 nadir category, the risk of cancer persisted at 5 years (RR = 2.33, 95% CI [1.27, 4.11], *p* = 0.0042) and at 10 years (RR = 1.77, 95% CI [1.04, 3.02], *p* = 0.0305).

## 4. Discussion

Based on a large, multi-center, multi-country cohort of CLWH across four countries in Eastern and Southern Africa, this study quantified the putative high incidence of cancers and identified risk factors for cancers in this population. The World age-standardized rate of 47.6 cancer cases per 100,000 person-years of CLWH in our study is approximately 3.5-fold higher than the estimated incidence of cancer in the general population of children 0–19 years in SSA of 13.5 per 100,000 person-years [26,27]. The relative incidence of cancer in CLWH was highest for KS. Compared to the World age-standardized rate of KS in children 0–14 years in SSA of 0.35 cases per 100,000 person-years, the World age-standardized rate of 36.6 in this cohort equates to a 104-fold higher incidence of KS. Similarly, the proportion of KS among cancers in these children was 13-fold greater than the approximately 5.91% seen in the general population of this age group in SSA [28]. The World age-standardized rate of lymphoma in these CLWH (8.94) was about 9-fold that of the general population of SSA children (0.98) [28]. Whereas it is suspected that the risk of lymphoma would vary by histological subtype such as NHL versus Hodgkin lymphoma, and, among NHL, Burkitt versus diffuse large B cell lymphoma [29,30], it was not feasible to determine these differences due to the low specificity of diagnoses and small number of lymphoma cases. The World age-standardized rate of leukemia was 68% lower in this cohort than in the general population of SSA children, suggesting lower incidence or poor detection [28].

This study identified delayed commencement of ART and severe CD4 lymphocyte nadir at any point during the life of CLWH as critical risk factors for developing cancer. The expansion of access to ART to CLWH in SSA over the past two decades has been a major public health intervention with wide-ranging benefits on the reduction of opportunistic infections and mortality in this population [31,32]. The association of severe CD4 suppression with cancer in the early years of HIV treatment of these children is consistent with the attribution of the benefits of ART in the prevention of opportunistic cancers to the restoration of the adaptive immune response [33]. Notably, this study found that an early initiation of ART, before age 2 years, was associated with significantly lower odds of developing cancer, independent of CD4 lymphocyte nadir and WHO HIV clinical stage. This finding supports the theory that other distortions of the adaptive immune system that are not measured by CD4 lymphocyte levels contribute to cancer predisposition in CLWH. Alternatively, it is possible that HIV directly predisposes to oncogenic events independent of its immunodysregulation mechanisms [34]. These findings further justify the current WHO recommendation to commence ART in children with HIV infection as soon as a diagnosis is made rather than after overt immunosuppression or advancing clinical stage. They also support the high value of screening mothers and newborns in high-risk populations for HIV infection [35,36].

Moreover, our results show that the relative risk of cancer in children with HIV is highest in the early years of life or infection. Although age and duration of infection are virtually indistinguishable in children because the vast majority acquire HIV through mother-to-child vertical transmission at birth, it is reassuring that the incidence density of cancer and impact of severe CD4 suppression or delayed ART waned over time with continued care and ART. However, there may be later peaks in HIV-associated cancer risk in adolescence or adulthood that require longer follow-up than reported in this cohort.

The overall patterns of cancer types and risk factors that we identified in this study are consistent with a records linkage study by Bohlius et al. (2016) that evaluated 29,348 pediatric HIV person-years and 24 incident cases of cancer in children with HIV infection in South Africa [37]. With 98,394 pediatric HIV person-years and 117 incident cancer cases, this study provides a more statistically robust perspective on incidence, types, and risk factors of cancer in children with HIV in SSA.

### 4.1. Limitations

Despite the large cohort size and access to comprehensive data in this cohort, the specificity of cancer diagnoses was a significant limitation, as it is in SSA generally [38,39,40]. The differences in rate of incident cancer cases in various BIPAI country centers as demonstrated in Table 1 is likely a result of differences in capability to suspect and diagnose cancers in CLWH. The case definition and classification of subtypes of cancer in this cohort were in some cases based on clinical history and examination only, without tissue histopathology (69.9% of incident cases had histopathological diagnosis in this cohort). Moreover, we found a differential tendency to more often diagnose KS clinically compared to lymphoma. Although the clinical diagnosis of KS is common practice in SSA with a positive predictive value of 77% in some studies, the current WHO standard is morphology and LANA immunohistochemistry of tissue biopsy [38,41]. In the case of lymphoma, this is a heterogenous group of cancers that particularly differ in immunophenotype depending on host immune status [42]. Unfortunately, none of the centers in this study had routine access to immunohistochemistry or other pathology tests to subtype the lymphomas. Thus, future studies should evaluate risk factors of each subtype differently. It was also not possible to evaluate the impact of protease inhibitors on risk of cancer in this cohort because only 5 of the 1605 children in the nested case-control sample ever received protease inhibitors as part of their ART, and none of the 5 developed cancer during the follow-up period.

This study used an arbitrary cut-off of three months from the time of enrollment in HIV care to cancer diagnosis to distinguish incident from prevalent cancer cases. It is not feasible to pinpoint the subclinical latent period of the types of cancers observed in this cohort, particularly because the children have other systemic illness and there are no evidence-based screening tests for these cancers. Cancers in HIV patients have been shown to occur prior to ART, and very early or long after commencing ART [43]. Furthermore, HIV viral load was rarely performed in this cohort. It would be a useful variable to examine the direct relationship between the level of viremia and cancer risk, regardless of immune function. Nevertheless, the study cohort is uniquely advantaged to study rare outcomes of pediatric HIV such as cancer because of the standardized care and EMR system in a multi-country setting in SSA. Future studies will focus on the prospective identification and ascertainment of cancer diagnoses using gold-standard criteria and assays of biomarkers of risk and screening of cancer in children with HIV infection on ART.

### 4.2. Conclusions

HIV is a strong predisposing factor to specific cancers in children in SSA. Antiretroviral therapy is a disease-modifying therapy that is expected to reduce the incidence of cancers in these children. This study quantifies the incidence of cancer in CLWH. This study also provides preliminary evidence for starting ART as early as possible and preventing severe CD4 lymphocyte suppression to reduce the risk of cancer in CLWH. This study suggests that the benefits of ART in cancer prevention go beyond restoration of CD4 lymphocyte suppression and are long-standing, at least through childhood, if therapy is continued. With the rapidly changing paradigm in funding of HIV care in SSA, it is critical that HIV treatment policies and programs such as routine maternal screening, prevention of mother-to-child transmission, monitoring of exposed newborns, and “diagnose and treat” are maintained to prevent complications such as cancer.

## Figures and Tables

**Figure 1 cancers-13-01379-f001:**
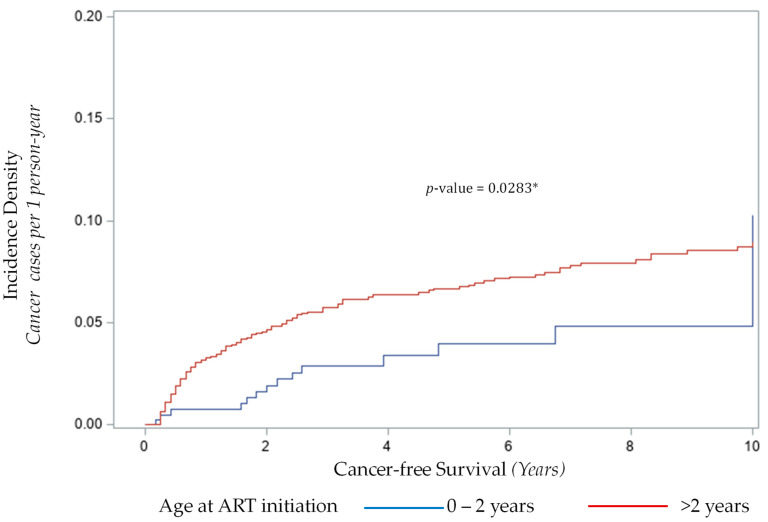
Incidence density function of cancer in children with HIV in BIPAI cohort by age at start of ART. *: *p* < 0.05.

**Figure 2 cancers-13-01379-f002:**
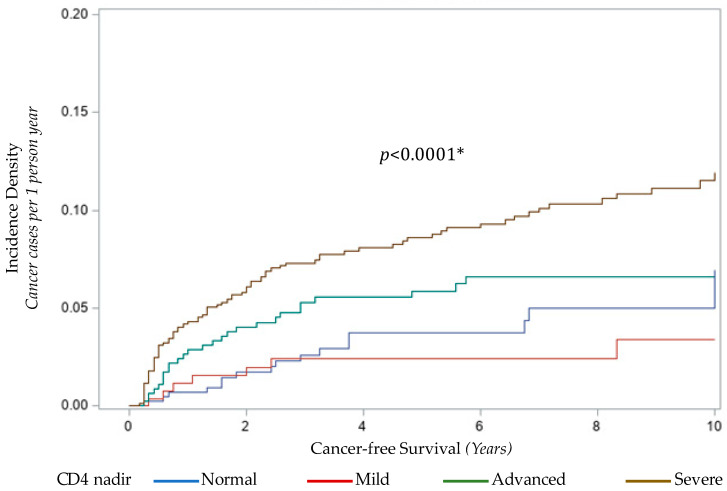
Incidence density function of cancer in children with HIV infection by CD4 Nadir during HIV care. *: *p* < 0.05.

**Table 1 cancers-13-01379-t001:** Demographic characteristics of nested incident cases and controls.

	Incident Cases (*n* = 117)	Controls (*n* = 1488)
Follow-up time (Years) (median, mean)	1.5, 2.28	1.50, 2.34
Age at enrollment (Years) (median, mean)	5.6, 6.9	4.9, 6.3
*Sex*		
Male (*n*, %)	70 (59.83%)	915 (61.49%)
Female (*n*, %)	47 (40.17%)	573 (38.51%)
*Country*		
Botswana (*n*, %)	8 (6.84%)	31 (2.08%)
Malawi (*n*, %)	45 (38.46%)	651 (43.75%)
Tanzania (*n*, %)	7 (5.98%)	0 (0.00%)
Uganda (*n*, %)	57 (48.72%)	806 (54.17%)
*Period of registration in HIV care*		
2004–2008 (*n*, %)	43 (36.75%)	486 (32.66%)
2008–2014 (*n*, %)	74 (63.25%)	1002 (67.34%)

**Table 2 cancers-13-01379-t002:** Distribution of subtypes of cancer among children with HIV infection in the Baylor College of Medicine International Pediatric AIDS Initiative (BIPAI) cohort.

Cancer Subtype	Prevalent Cases *n* = 310	Incident Cases *n* = 117	Crude Incidence Density ^1^ (95% CI)	Adjusted ^2^ Incidence Density ^1^ (95% CI)
Kaposi Sarcoma	226 (72.90%)	90 (76.92%)	91.5 (74.4, 112.5)	36.6 (26.5, 50.5)
Non-Hodgkin Lymphoma	18 (5.80%)	17 (14.53%)	17.3 (10.7, 27.8)	8.9 (4.7, 17.0)
Hodgkin Lymphoma	2 (0.65%)	5 (4.27%)	5.1 (2.1, 12.2)	
Leukemia	1 (0.32%)	1 (0.85%)	1.0 (0.2, 5.7)	0.4 (0.0, 4.6)
Solid Tumors ^3^	5 (1.61%)	4 (3.42%)	4.1 (1.5, 10.8)	N/A

^1^ Incidence density units are “Per 100,000 person-years.” ^2^ Adjusted to the World age-standardized rate. ^3^ Solid Tumors included brain tumor, fibrosarcoma, nasopharyngeal carcinoma, osteosarcoma, retinoblastoma, rhabdomyosarcoma.

**Table 3 cancers-13-01379-t003:** Risk factors for cancer in children with HIV infection in the BIPAI cohort.

Risk Variable	HIV and Cancer *n* = 117	HIV No Cancer *n* = 1488	Odds Ratio (95% CI)	Adjusted * Odds Ratio (95% CI)
*Age at ART initiation*				
0–2 years	13 (11.11%)	377 (25.34%)	-	-
>2 years	104 (88.89%)	1111 (74.66%)	2.71 (1.51, 4.89) **	2.84 (1.57, 5.13) **
*CD4 nadir*				
Normal	15 (12.82%)	223 (14.99%)	-	-
Mild	7 (5.98%)	203 (13.64%)	0.51 (0.19, 1.24)	0.52 (0.21, 1.31)
Advanced	23 (19.66%)	306 (20.56%)	1.12 (0.58, 2.23)	1.12 (0.57, 2.19)
Severe	72 (61.54%)	756 (50.81%)	1.42 (0.82, 2.61)	1.44 (0.81, 2.57)
*WHO Stage at enrollment ****				N/A
I–III	56 (73.68%)	912 (84.92%)	*-*	
IV	20 (26.32%)	162 (15.08%)	2.01 (1.18, 3.44) **	

* Adjusted for WHO stage at time of enrollment in care. ** *p* < 0.05. *** 41 cases and 414 controls had missing WHO clinical stage data.

## Data Availability

Data is not publicly available and is the property of each respective Ministry of Health and the authors do not have approval to share these data outside of our analyses.

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
