# Peer review of "Association between Antiretroviral Therapy and Cancers among Children Living with HIV in Sub-Saharan Africa"

_cancers, 2021, doi:10.3390/cancers13061379_

Round 1

Reviewer 1 Report

The topic is challenging on many aspects which make the data presented in this manuscript are very interesting and of potential interest for the readership of the journal.

My main concern is some missing information that impair the robustness and the impact of the conclusions.

  • Section 2.1 “Study design and study population”. It is difficult to estimate the representativeness of the population analysed in the manuscript with the description given in this section. Can the authors elaborate on this and discussed this point in the study limitation section?
  • Table 1. The rate of incident cases are different from one country to the others. Can it be discussed at some point?
  • Table 3. The number of confounding variables is rather limited. Can “sex”, “site”, “type of ARV regimen”, “time to CD4 normalisation” (no viral load available), “infant prophylaxis during first year of life” be included in the model?

Minor points:

  • L84 in the introduction. Only PI are listed as putative antitumor agent. Other ARV classes could be added as they are undergoing repositioning in different cancer trials. It is noteworthy that AZT was first selected as an antitumor agent during the 60ies.
  • L49: “HIV enrolment” sounds inappropriate and difficult to understand.

Author Response

Section 2.1 “Study design and study population”. It is difficult to estimate the representativeness of the population analyzed in the manuscript with the description given in this section. Can the authors elaborate on this and discuss this point in the study limitation section?

Thank you for this suggestion. We have added text in Lines 106 – 111 to elaborate on the representativeness of the study population, in three contexts: 1) Geographically i.e. representative of sub-Sahara Africa; 2) Diagnosis and, 3) Treatment practices of pediatric HIV, i.e. study population includes both earlier (about 2004 – 2008) practices where children were predominantly diagnosed after presenting with AIDS-defining illnesses and started on ART after evidence of severe immunosuppression, and the later (about 2009 – 2014) when children were diagnosed with HIV through early infant screening programs and started on ART immediately.   

Table 1. The rate of incident cases are different from one country to the others. Can it be discussed at some point?

We discuss this difference in the Limitations Section 4.1., Lines 264 – 266. The difference in incident cancer rates is likely to be a result of clinical and laboratory capabilities to diagnose cancer in children with HIV infection in these country centers. 

Table 3. The number of confounding variables is rather limited. Can “sex”, “site”, “type of ARV regimen”, “time to CD4 normalization” (no viral load available), “infant prophylaxis during first year of life” be included in the model?

Thank you! Variables that were considered for adjustment in the model, had to meet certain criteria: 1) Clinical or biological importance; 2) Data available; 3) Association with cancer on univariate analysis. Sex did not meet the association criterion on univariate analysis. Site/country was eliminated because of low clinical and biological significance and risk of differential misclassification between countries due to differences in cancer suspicion or diagnosis capability. Type of ARV regimen was eliminated because of very small sample size on protease inhibitors. Time to CD4 normalization was deemed to have a high likelihood of misclassification bias because CD4 counts were not performed at regular and standardized/same intervals for all children. Age at start of ART was a key independent variable as indicated in Table 3 and would correlate very closely with “infant prophylaxis during the first year of life.”

L84 in the introduction. Only PI are listed as putative antitumor agent. Other ARV classes could be added as they are undergoing repositioning in different cancer trials. It is noteworthy that AZT was first selected as an antitumor agent during the 60ies.

Clarified that other ARV classes may have anti-tumor effects in Line 84 – 85.

L49: “HIV enrolment” sounds inappropriate and difficult to understand

Restructured phrase to read “…. enrolment for HIV care” instead of “…. HIV enrolment”

Reviewer 2 Report

The Authors reported a very well conducted and written cohort study exploring the subtypes, incidence and risk factor of cancers in CLWH in SSA. The manuscript represents a very comprehensive study on this topic, with very original data and without similar data in literature. The study has some limitations, due to the specific nature of the study itself, but the Authors clearly reported them in the specific session. 

Author Response

We thank the reviewer for highlighting the significance and importance of this manuscript. 

Reviewer 3 Report

The authors present results from a nested case-control study of children 0-18 years from 2004 – 2014 at five centers in sub-Saharian Africa (SSA) to determine the incidence and risk factors for cancer among children living with HIV (CLWH). There were three major findings:
First, the adjusted incidence density of all cancers, Kaposi sarcoma, and lymphoma were 47.6, 36.6 and 8.94 per 100.00 person years, respectively.
Second, delayed ART until 2 years of age was associated with cancer (OR 2.71) after adjusting for WHO clinical stage at time of HIV enrolment, and
third, the relative risk of cancer associated with severe CD4 suppression was 6.19, 2.33 and 1.77 at one, five and 10 years of ART, respectively.

The majority of HIV-infected children live in Sub-Saharan-Africa, but cancer studies in HIV-infected children from this region are scarce. Thus, the present work is worth reporting. Overall, the paper is well written, the methods used are appropriate, and limitations of the study are adequately addressed.

I would only suggest that the authors also discuss recent findings from a South African study in which similar results were reported by Bohlius et al: Incidence of AIDS-defining and Other Cancers in HIV-positive Children in South Africa: Record Linkage Study. Pediatr Infect Dis J 2016;35(6):e164-70.

Author Response

I would only suggest that the authors also discuss recent findings from a South African study in which similar results were reported by Bohlius et al: Incidence of AIDS-defining and Other Cancers in HIV-positive Children in South Africa: Record Linkage Study. Pediatr Infect Dis J 2016;35(6):e164-70.

Thank you for pointing us to this publication. We discuss the similarities in overall findings and the differences and value added by our manuscript (Lines 261 – 266).

Round 2

Reviewer 1 Report

I recommend that the answer to point 3 be inserted in section 2.4 "Study variables and definitions". I believe that if I have questioned the non-use of these variables in the model, other readers will do so. It is preferred to include these justifications in the M&M section.

Author Response

 In response to the suggestion “I recommend that the answer to point 3 be inserted in section 2.4 "Study variables and definitions". I believe that if I have questioned the non-use of these variables in the model, other readers will do so. It is preferred to include these justifications in the M&M section.”; We have added the explanation to the manuscript, (see lines 135 – 143) i.e., that variables that were considered for adjustment in the model, had to meet certain criteria: 1) Clinical or biological importance; 2) Data available; 3) Association with cancer on univariate analysis. Sex did not meet the association criterion on univariate analysis. Site/country was eliminated because of low clinical and biological significance and risk of differential misclassification between countries due to differences in cancer suspicion or diagnosis capability. Type of ARV regimen was eliminated because of very small sample size on protease inhibitors. Time to CD4 normalization was deemed to have a high likelihood of misclassification bias because CD4 counts were not performed at regular and standardized/same intervals for all children.